# A Biomechanical Waist Comfort Model for Manual Material Lifting

**DOI:** 10.3390/ijerph17165948

**Published:** 2020-08-16

**Authors:** Yongbao Zhang, Jinjing Ke, Xiang Wu, Xiaowei Luo

**Affiliations:** 1School of Engineering and Technology, China University of Geosciences (Beijing), Beijing 100083, China; 2102180103@cugb.edu.cn; 2Department of Architecture and Civil Engineering, City University of Hong Kong, Hong Kong 999077, China; jingke4-c@my.cityu.edu.hk (J.K.); xiaowluo@cityu.edu.hk (X.L.)

**Keywords:** manual material handling, waist comfort level, biomechanics

## Abstract

Low back pain (LBP) is a common disorder that affects the working population worldwide. LBP causes more disability than any other conditions all around the world. Most existing studies focus on the occupational physical factors in association with LBP, while few focus on individual factors, especially the lack of quantitative calculation of waist comfort in biomechanics. Based on the physical statistics of Chinese men, this research used human posture analysis (HPA) to establish the waist strength formula and analyzed the waist strength during a manual material handling. It also explored the influence of weight and height of lifting objects on the L5-S1 spinal load. On this basis, a waist comfort model was proposed in combination with the recommended weight limit (*RWL*) recommended by NIOSH, and the parameter selection and waist comfort value were verified by Jack simulation software. The results show that pulling force of the Erector Spinae of the waist is closely related to the weight and lifting height of the object. Parameter verification and Jack software simulation results show that the force of L5-S1 is less than 3400 N, which proves that the waist force under this posture is acceptable. The developed waist comfort model can be applied to evaluate work risk, to adjust working intensity and powered exoskeleton design, aiming to decrease the prevalence of LBP.

## 1. Introduction

Low back pain (LBP) is a major public health concern for all age groups. LBP can progress from a chronic pain condition to a more complex condition, resulting in adverse impacts on spinal muscles, nerves, bones, and discs in the lower back region [1]. LBP accounts for a large amount of workers’ compensation in all countries and affects the overall productivity of the society and the life quality of human [2]. LBP is considered as a major disease of Musculoskeletal disorders (MSDs), its point prevalence was up to 9.4% globally, and it was identified as the leading cause for disability according to the 2010 Global Burden of Disease Study [3]. Therefore, LBP has attracted considerable attention both from the public health practitioners and the academia in recent years. Sweden, Germany, the United States and International Labor Organization (ILO) have confirmed that LBP falls into the scope of occupational impairment, and the cost of medical treatment for LBP is estimated to exceed 100 billion dollars per year in the United States [4]. Therefore, prevention of LBP is more proactive and cost-effective compared to the treatment of LBP.

The most common form of low back pain is nonspecific, affecting almost all age groups [5]. However, in different occupational backgrounds, workers’ repeated bending of the spine, frequent heavy lifting, manual material handling and awkward body postures are still important factors that cause MSDs of the waist [6]. Back compressive force on the spine is believed to be a contributor to low-back pain and injury, especially at the L5/S1 intervertebral disc [7]. Frequent heavy lifting, awkward postures, and manual lifting are commonly observed for workers in the construction, mining and shipbuilding industries [8]. The heavy lifting tasks and repetitive bending have been considered as the primary causes of back muscle fatigue, spinal injury and back injury [9]. In order to help workers complete the lifting task, cranes, wearable exoskeletons [10] and clothing and other equipment have been applied in the production line [11]. However, manual lifting cannot be entirely replaced by machinery, especially for the tasks involving small objects, due to the high cost and learning curve of those machinery-assisted measures [12]. Therefore, it is still very necessary to research and evaluate the waist stress during manual lifting.

Biomechanical models are used by ergonomics to estimate the lumbar intervertebral compression forces experienced during infrequent occupational lifting tasks. The estimates are then compared to selected tissue tolerance values to assess the risk of low-back injury for a given load and posture [13]. In terms of lifting activities, various spinal load measurement methods have been developed to evaluate spinal load and the risk of being susceptible to LBP. Evaluation methods include self-reporting methods, direct measurement methods (such as 3D static strength prediction program (3DSSPP), lumbar motion monitor (LMM), Electromyography (EMG) sensors based on biomechanical analysis, and video or image sensors to capture the waist sports-based camera technology [14]. Among these methods, the biomechanical load analysis is objective and relatively precise without subjective evaluation bias and angle errors which are common in video capture systems. In recent decades, researchers developed biomechanical models to quantify spinal load during manual materials’ lifting, including static models based on the postures analysis [15], dynamic models considering movements [6], cumulative spinal load (CSL) calculation and 3D biomechanical models involve twisting. An early study refined the static sagittal-plane (SSP) biomechanical model conducted by Morris [16], which simplified the human body as a series of seven links and computed corresponding forces and torques during simulated manual materials carrying tasks [17]. The National Institute of Occupational Safety and Health (NIOSH) proposed a formula to estimate lifting loads in the sagittal plane, and later, in 1991, revised the recommended lifting formula, which was expanded to apply for asymmetrical lifting tasks. The revised formula considered several variables, including the standard lifting location, load constants, horizontal and vertical distance of the load from the spine, asymmetric multiplier, frequency and coupling multipliers [18]. Additionally, the 1991 committee generally identified 3400 N on the lumbar L5-S1 as the maximum compressive force criterion [3]. The Lifting Index (LI) is the ratio of the load lifted to the recommended weight limit. In theory, the magnitude of the LI may be used as a gauge to estimate the percentage of the workforce that is likely to be at risk for developing lifting-related low back pain [19]. Waters et al. believed that LI is related to waist risk [18]. Garg pointed out that when IL is lower than 1, the task is considered safe. An IL greater than 1 means that it may cause risks to some workers, and as it increases, the risk of the task will also increase, even if the relationship between LI and the risk of excessive sports injury is not linear [20]. Moreover, researchers identified that higher probability of low back injuries exists when the any of the following lifting conditions is met [21]: fast lifting pace, increasing lifting height from the ground, increasing weights of objects [22]. In addition, a recent study demonstrated that different risk subjects to LBP exist between males and females in the same task [23]. A variety of lifting conditions were investigated for manual lifting tasks, and different quantitative methods of lumbar spine load were proposed. However, formulas and models based on authoritative institutions such as NIOSH are not fully applicable to Asian studies. Therefore, it is still of positive significance to establish a waist static balance model to explore how to reduce the risk of waist and back injuries during Chinese male handling operations and to improve waist comfort.

Human comfort is an intuitive feeling of humans, affected by environmental, physiological and psychological factors. Human comfort can be considered as a reaction to the surrounding environment. As such, discomfort levels related to lifting tasks were usually obtained through subjective recalls or surveys [24], one of which is ratings of perceived discomfort (RPDs) occurring in six major body areas (neck, shoulders, upper back, lower back, hips/thighs, and knees) [25]. From the physiological point of view, many factors cause discomfort, including joint force, muscle load, and vibration. The existing subjective measure of discomfort level is greatly affected by individuals and might cause bias. Therefore, an objective model of waist comfort model is needed. To address this gap, the objective of this study is to build a waist comfort model to quantify comfort levels while performing lifting activities.

Numerous studies have investigated the effects of different lifting conditions, such as the speed of lifting, the weight of an object, and the frequency of lifting, but most of them use questionnaires to understand the waist comfort of subjects, and there is a lack of quantitative research. Therefore, from the perspective of establishing a static biomechanical model, it is still very meaningful to explore the influence of factors such as the height and weight of people in different regions and the weight and height of lifting objects on the strength and comfort of the spine. It is necessary to take individual body differences into account while workers perform the same task or the same load criteria [26]. Therefore, based on physical statistics of Chinese men, the present study aims to figure out a more reasonable and scientific relationship between human height and weight and spinal load on L5-S1, and to establish a biomechanical waist comfort model. Besides this, the biomechanical simulation software called Jack will be applied to verify this comfort model, thus providing workers’ manual lifting with a reliable estimating model. On the one hand, the established static mechanics model can not only improve the current ergonomic suggestions, but also provide a reference for future research on dynamic work intensity and work posture adjustment. On the other hand, the calculation of waist force and comfort can be regarded as an indicator for future wearable exoskeleton designs. The comfort model is also expected to improve the working status and health concern of workers, foster productivity and create benefits for the enterprise, which can reduce the incidence of LBP and expenses on work-related LBP disorders.

## 2. Materials and Methods

To analyze the waist force and develop the comfort model, the human posture analysis (HPA) method was adopted for this study. Since the stoop posture is widely used in manual lifting, the whole process is based on the stoop posture analysis in the sagittal plane, which belongs to the static spinal load without torsion. First, the waist force formula on L5-S1 was conducted, and the relationship among human weight and height, object’ weight and height and the L5-S1 spinal load were obtained. Based on the waist force formula, the waist comfort model was proposed, combined with *RWL* recommended by NIOSH.

### 2.1. Waist Force Model

#### 2.1.1. Selection of Waist Force Loading Point

Uneven force or injury of the Erector Spinae can cause waist discomfort and even musculoskeletal disorders [27]. Some studies have shown that main force is made by low back and the L5-S1 lumbar disc, the disc is located between the fifth lumbar vertebra and the first sacral vertebra [28], which is the most fragile part in handling work [29,30]. Considerable compressive stress will pose on the spine of the L5-S1 region and long-time compression, and sudden twisting or bending will lead to intervertebral disease during lifting activities. To quantify the force of the Erector Spinae when workers are at stoop posture, it is necessary to build the stress model which is close to actual work situation on the basis of the biomechanics of the human body. Hence, this paper regards L5-S1 as the force loading point of the Erector Spinae. Since the biological system of the human body is extremely complicated, simplified and abstracted methods are required to facilitate the study by biomechanical means. The human body is considered as a rigid body because the deformation of the vertebral body under a load is minimal, so as to be ignored compared with the other parts of the vertebral and ligaments. In a static state, the force of the objects must satisfy two conditions: (1) the external force of the action on the object is zero, and (2) the sum of the torque of each external force on the object is zero. These two conditions play a vital role in the biomechanical model. The stoop posture is defined as keeping the legs straight and bending the waist forward and downward in the sagittal plane, so the stoop posture in this paper refers to bend in the sagittal plane and the torsion is not considered.

#### 2.1.2. The Relationship between F (Force of Erector Spinae) and G (Gravity of the Moving Object)

Assuming that a person remains stationary at a stoop posture, the human body satisfies the static equilibrium condition shown in Formula (1): ∑*M* = 0. L5-S1 can be regarded as a fulcrum when only the gravity *G* and the pulling force *F* of the vertical ridge muscle are considered during manual lifting. The body part above L5-S1 is simplified as a whole set, and its gravity (vertically downward) multiplies the distance of L5-S1 to the simplified center of gravity creates a clockwise torque. The product of the object’s gravity and distance from its center to L5-S1 creates an additional clockwise torque. Force *F* produced by the vertical ridge muscle multiplies the distance from the force to the L5-S1 creates a counterclockwise torque [13]. In this way, a simple force model based on the torque balance is developed, as shown in Figure 1a.
(1)∑M=0
(2)$$M=F\cdot L=G\cdot L_{1}+F_{1}\cdot L_{2}$$

Based on Figure 1a, the relationship among *M*, *F* and *G* were obtained in Formula (2), where M is the moment of force (unit: N*m); *F* is the force of Erector Spinae (unit: N); *L* is the distance from the Erector Spinae to L5-S1, about 5 cm (unit: cm); *F*_1_ is the total mass of the body above the lumbar spine, including the head, trunk, and arms (unit: N); *L*_2_ is the horizontal distance between *F*_1_ and L5-S1 (unit: cm); *G* is the gravity of the object (unit: N); *L*_1_ is the horizontal distance from *G* to L5-S1 (unit: cm) [7,16].

When investigating the relationship between *F* and *G*, it is stipulated that other physical quantities have fixed values. Therefore, M has a linear functional relationship with *G*, and *F* produced by the erector muscle has a linear function relationship with *G*, so *F* increases with the increase in gravity *G*.

#### 2.1.3. The Relationship between F (Force of Erector Spinae) and h (Height of the Moving Object)

The L5-S1 still considered as the fulcrum, and the body above the L5-S1 can rotate around a fulcrum in the sagittal plane if the height of the object changes. The change in the height (*h*) of the object directly affects the value of the waist angle *α*. The simplified force model based on the torque balance is built in reference to Figure 1b. Based on the static equilibrium condition, Formula (3) can be derived.
(3)$$M=F\cdot L=G\cdot L_{1}+F_{2}\cdot L_{2}+F_{3}\cdot L_{3}$$

From Figure 1b, Formulas (4) and (6) can be derived.
(4)$$\cos\alpha =\frac{H_{2}+h-H_{1}}{L}$$
(5)$$\sin\alpha =\sqrt{1-\cos^{2}\alpha }=\sqrt{1-\frac{{\left(h+H_{2}-H_{1}\right)}^{2}}{L^{2}}}$$
(6)$$L_{1}=L_{3}=L\cdot \sin\alpha ,L_{2}=k\cdot L\cdot \sin\alpha$$

When substituting *L*_1_, *L*_2_, *L*_3_ in (3) with Formula (6) and in (6) with Formulas (4) and (5), Formula (3) becomes Formulas (7) and (8)
(7)$$M=(G+k\cdot F_{2}+F_{3})\cdot L\cdot \sqrt{1-\frac{{\left(h+H_{2}-H_{1}\right)}^{2}}{L^{2}}}$$
(8)$${\left(\frac{M}{a}\right)}^{2}+{\left(\frac{h+c}{b}\right)}^{2}=1$$

When discussing the relationship between F and h, it is stipulated that other physical quantities be fixed values. Let (*G* + *k* × *F*_2_ + *F*_3_) × *L* = *a*, *L* = *b*, *H*_2_ − *H*_1_ = *c*, (*a*, *b*, *c*, are all constants). In Figure 1b, A is L5-S1, B is the gravity center of the trunk and head, C is the acromion point, and D is the center of gravity of the arm. Arm vertical downward and AC is trunk length L, set AB = *k* × *L*. *G* is the gravity of carrying object (unit: N). *L*_1_ is the horizontal distance from *G* to L5–S1 (unit: cm). *F*_2_ is the total mass of the trunk and the head of the body above the lumbar spine (unit: N). *L*_2_ is the horizontal distance from *F*_2_ to L5-S1 (unit: cm). *L*_3_ is the horizontal distance from *F*_3_ to L5-S1 (unit: cm). *h* is the vertical distance of the center of gravity *G* from the ground (unit: cm). *H*_1_ is the vertical distance from L5-S1 to the ground (unit: cm). *H*_2_ is arm length (unit: cm). *α* is the trunk inclination angle (0 < *α* ≤ 90) (unit: degree).

*F* and *h* have the elliptic function relation of quadratic function as *M* and *h*. Because 0 < *α* ≤ 90 and the image are in the first quadrant of the coordinate system, thus *F* monotonically decreases with h.

#### 2.1.4. Static Equilibrium Model of the Waist

During a manual lifting, the vertical ridge muscle is stretched to produce pulling force. The body is supported by the lumbar intervertebral disc L5-S1 and the body part above the L5-S1 is equivalent to a lever from the perspective of biomechanical. The weight of the body above L5-S1 and workload are the resistance of the lever, and the pulling force produced by the vertical ridge muscle is the primary motive force. Since the power arm is less than the resistance arm, the body is a laborious lever. Based on the relationship mentioned, the body parts above L5-S1 are divided into the head and neck, the trunk and the arm. The mechanics model of the waist is built as shown in Figure 1c and the following formulas were obtained
(9)$$M_{F}+F_{4}L_{4}=F_{1}L_{1}+F_{2}L_{2}+F_{3}L_{3}+GL_{2}$$
(10)$$M_{F}+F_{4}L_{4}=F_{5}L_{1}+F_{3}L_{2}+F_{6}L_{3}+GL_{2}$$
(11)$$M_{F}+F_{4}L_{4}=j_{1}M\cdot k_{1}H\sin\alpha +j_{2}M\cdot k_{2}H\sin\alpha +j_{3}M\cdot k_{3}H\sin\alpha +G\cdot k_{2}H\sin\alpha$$
(12)$$M_{F}+F_{4}L_{4}=M\cdot \left(j_{1}\cdot k_{1}+j_{2}\cdot k_{2}+j_{3}\cdot k_{3}+\frac{G}{M}\cdot k_{2}\right)\cdot H\sin\alpha$$
(13)$$M_{F}+F_{4}L_{4}=MH\cdot \left(j_{1}\cdot k_{1}+j_{2}\cdot k_{2}+j_{3}\cdot k_{3}+\frac{G}{M}\cdot k_{2}\right)\cdot \sqrt{1-{\left(\frac{h}{k_{2}H}+\frac{b-a}{k_{2}}\right)}^{2}}$$
where A is L5-S1, B is the center of gravity of the trunk part of the body above the lumbar spine, C is the acromion point, and D is the center of gravity of the head. *G* is the gravity of the object (unit: N). *F*_5_ is the gravity of the head (unit: N), and *L*_1_ is the horizontal distance from the center of gravity of the head D to L5-S1 (unit: cm). *F*_3_ is the gravity of the arm (unit: N), *L*_2_ is the horizontal distance between acromion point C and object *G* to L5-S1 (unit: cm). *F*_6_ is the body gravity of the trunk above the lumbar spine (unit: N), *L*_3_ is the horizontal distance between from B to L5-S1 (unit: cm). *h* is the vertical distance from *G* to the ground (unit: cm). *H*_1_ is L5-S1 vertical distance from the ground (unit: cm). *H*_2_ is the length of the arm (unit: cm;); *F*_4_ is the force formed by intra-abdominal pressure (unit: N), and *L*_4_ is the distance from *F*_4_ to L5-S1 (unit: cm). *α* is the trunk inclination angle (0 < *α* ≤ 90) (unit: degree). *H* is human height (unit: cm), assume: AD = *k*_1_ × *H*, AC = *k*_2_ × *H*, AB = *k*_3_ × *H*, *H*_1_ = *a* × *H*, *H*_2_ = *b*. *M* is human gravity (unit: N), assume: *F*_5_ = *j*_1_ × *M*, *F*_3_ = *j*_2_ × *M*, *F*_6_ = *j*_3_ × *M*. (*k*_1_, *k*_2_, *k*_3_, *j*_1_, *j*_2_, and *j*_3_ are all constants and are fixed values).

According to the Formula (13), the variables are *M*, *H*, *G*, *h*, and the rest are constants. It is known that the torque *MF* is related to the values of the above variables, so the force *F* produced by the vertical ridge muscle is also related to these variables. To obtain the specific value of *F*, it is necessary to determine the human data *M*, *H*, constants *a*, *b*, proportional parameter *k*_1_, *k*_2_, *k*_3_, *j*_1_, *j*_2_, *j*_3_.

#### 2.1.5. Parameters Selection

The parameters in this paper are with reference to adult human body inertial parameters (GB/T 17245-2004) and main body size in men (GB10000-88) of China, and the anthropometric data from relevant books and documents as shown in Table 1.

##### Determine *M* and *H*

According to the inertial parameters of the adult human body (male): the whole centroid position is 734.2 mm (the starting point of the body center with the head vertex), and the relative position of the whole centroid is 43.8%. It can be calculated that the standard height is *H* = 734.2/43.8% = 1676 mm = 168 cm. The fiftieth percentile of male Chinese adult human body height is 1678 mm = 168 cm, so *H* is 168 cm, and the corresponding weight is 59 kg, M = 59 × 9.8 = 578 N.

##### Determine the Constant *a*, *b*

The height *H* is 168 cm, *H*_1_ = *a* × *H*, *H*_2_ = *b*. To obtain a and b, H_1_ and H_2_ must be determined. The height of the superior anterior spine is L5-S1 because the height of L5-S1 is similar to that of the anterior superior iliac spine. The height of the anterior superior iliac spine is 254.5/(1 − 45.3%) + 224.1/(1 − 39.3%) + 38.2/(1 − 38.2%) = 909 mm = 91 cm, and the length of the thigh is 465 mm (the proximal thigh point is the anterior upper iliac spinous point, the far side is the tibial point) in adult human body size, and the tibial point is high 444 mm. The height of the tibial point can be calculated as 909 mm. For the convenience of calculation, *H*_1_ = 90 cm and *a* = 90/168 = 0.54 were taken. The fiftieth percentile of male (18~60 years) shoulder height and high hand function in Chinese adult human body size is 1367 and 741 mm respectively, so *H*_2_ = 1367 − 741 = 626 mm = 62.6 cm, b = 62.6/168 = 0.37.

##### Determine the Proportions *j*_1_, *j*_2_ and *j*_3_ on Body Weight

As shown in Table 1, the relative mass of the head and neck is 8.62%, that is *j*_1_ = 8.62%*M*, the relative mass of the upper arm, forearm and hand are 2.43%*M*, 1.25%*M*, and 0.64%*M*, that is *j*_2_ = (2.43% + 1.25% + 0.64%) × 2*M* = 8.64%*M*. According to Morris et al.’s study [16], the gravity of the trunk part above L5-S1 is *F*_3_ = 30%*M*, so *j*_3_ = 30.00%*M*.

##### Determine the Proportionality Coefficient *k*_1_, *k*_2_, *k*_3_

The distance from head to L5-S1 is L, equaling to the height subtract high anterior iliac spine point. Therefore, *L* = 168 − 91 = 77 cm. Reference to the center of the head and neck in Table 1, it is located at 117.8 mm from the top of the head. The distance from the head to the head to the L5-S1, AD = *L* − 117.8 mm = 65 cm, and *k*_1_ = 0.39 by AD = *k*_1_ × *H*. From the acromion point to L5-S1, AC equals acromion high subtract high iliac superior spine point. According to the adult human body size, the shoulder height is 1367 mm. As the upper iliac upper spine is 909 mm high, AC = 1367 − 909 = 458 mm = 46 cm, AC = *k*_2_ × *H* = 0.27. The distance between the lower chest points and the L5-S1 is defined as the length of the lower trunk (the proximal point of the upper trunk is the point of the cervical spine, and the distal point is the lower point of the chest), and the middle point of the above torso and the lower trunk is the center of gravity. According to Luo et al., AB= (height-head and neck-length-superior iliac spine point high)/2 [31], so the length of the head and neck is 117.8/46.9% = 251 mm = 25 cm, AB = (168 – 25 – 91)/2 = 52/2 = 26 cm, AB = *k*_3_ × *H* = 0.15.

##### Other Data

Morris and others identified *L*_4_ = 114.3 mm = 6.80%*H*, *F*_4_ = 1%*M*. The distance between *F* and L5-S1 is about 50 mm, that is, 2.98%*H* [16]. These data are substituted into Formula (13), and the resulting Formula (14) is as follows.
(14)$$F=\left(3.42M+9.06G\right)\cdot \sqrt{1-{\left(\frac{0.37h}{H}-0.63\right)}^{2}}$$

#### 2.1.6. Analysis of Static Equilibrium

The Formula (14) shows that the force of the vertical ridge muscle is related to the weight and the height of the moving objects, which is also associated with the height and weight of the moving workers. The force of the L5-S1 varies among different workers under the same handling tasks. In the discussion of the relationship between *F* and *G*, the torso inclination angle *α* is a definite value. It is known from the mass is 59 kg, that is, *M* = 578 *N*. Therefore
(15)F=(1976.76+9.06G)sinα

The conclusion that *F* is proportional to *G* satisfies the Formula (15). The maximum torque value of the lower waist *MF* significantly increases with the increasing load of lifting weight when the distance from *F* to L5-S1 is fixed. When *G* = 0 *N*
(16)F=3.42M∗sinα

When the relationship between *F* and *h* is discussed, the range of the bending angle is 0 < *α* ≤ 90 (unit: degree), 0 ≤ cos*α* < 1, that is, 0 ≤ 3.70 *h*/*H −* 0.63 < 1. When the value of *H* is 168 cm, 28.6 cm ≤ *h* < 74.0 cm.

The main content of this chapter is focused on analyzing the biomechanical aspects of the force of the waist in the handling work. A simplified model of human body mechanics is set up and the relationship between *F* and *h*, *G* is obtained. Besides these, the formula derivation and the parameters selection are given, and the main factors that affect the force of the waist are identified.

### 2.2. Waist Comfort Model

#### 2.2.1. Build Comfort Model Based on the Recommended Weight Limit (*RWL*) and Lift Index (LI)

Comfort is a feeling of people. The degree of comfort can be summarized as follows: (1) Comfort is a subjective feeling of a person, which is relative to the degree of discomfort. (2) The factors that affect comfort degree include physiological factors and psychological factors. The study of comfort must start from two aspects: physiology and psychology. In this chapter, from the physiological aspects, combined with biomechanics knowledge, the waist comfort of workers during lifting is expected to be quantified.

Labor intensity refers to the degree of labor consumption during a period. The classification of physical labor intensity is of great significance to the dynamic analysis of working ability, the measurement of fatigue, the reduction in fatigue and the improvement in the working mode [32]. In 1983, the national standard bureau announced the standard of manual labor intensity GB3869-83. In this standard, the labor intensity index (the function of two factors of energy metabolism rate and labor time ratio) was put forward to classify manual labor intensity. In 1997, the standard was updated to GB386997, in which the labor intensity index is further differentiated in the sex coefficient and the coefficient of work. According to the good linear relationship between the energy metabolism rate and the heart rate, Luger, T used the relative heart rate as the explanatory variable, considering factors such as sex, environment, age and other factors to establish the labor intensity evaluation model [33]. This section tries to establish the waist comfort model in the manual handling and calculate the comfort degree by quantitative method. The value of comfort can be viewed as an evaluation reference of labor intensity. Yang Feng describes the astronaut’s discomfort level by the relative relationship between the torque used by the joint and the maximum available torque of the joint [6], which is defined as the ratio of the actual torque tau and the maximum force moment max of the joints, like the followings
(17)$$\alpha =1-\beta ,\beta =\frac{\tau }{\tau_{\max}};\alpha =1-\frac{\tau }{\tau_{\max}}$$

Waist comfort model is defined as
(18)$$C=1-\mu C=1-\frac{F}{F(RWL)}$$
where *F* is pulling force generated by erector muscle in handling, which can be calculated by Formula (14). *RWL* is the recommended moving value by the NIOSH; *F* (*RWL*) is the pulling force of the erector muscle when the weight is *RWL* at the same height; *μC* is waist discomfort is defined as the ratio of *F* to *F* (*RWL*), that is, *F*/*F* (*RWL*). *C* is waist comfort is defined as 1-discomfort, 1 − *μC*. According to Formula (14), *F* produced by the erector muscle is the function of height *H*, human gravity *M*, lifting height *h* and moving weight *G*. Thus, the *F* (*RWL*) can be calculated if the recommended weight limit (*RWL*) and other human parameters can be determined. It is known from the above that the tension *F* of the vertical ridge muscle is the monotonous increasing function of the gravity *G* of the object, and the relation between the *F* and *F* (*RWL*) can be determined by the relation between *G* and the *RWL*.

#### 2.2.2. Recommended Weight Limit (*RWL*)

NIOSH put forward a lifting limit formula of the human body, combining the disciplines of biomechanics, psychophysics, and psychology. In 1991, the lifting formula was trimmed and *RWL* and lifting index (the ratio of actual weight to the recommended weight) were added to the formula for the evaluation of manual lifting task. The *RWL* means that for specific working conditions, almost all healthy workers can work well enough for a long time (for example, for 8 h) without causing an increase in the weight of the associated lower back pain risk [18]. NIOSH revised the *RWL* lifting formula in 1991 as follows
(19)$$RWL=23kg*(\frac{25}{H})*(1-0.003\left|V-75\right|)*\left(0.82+\frac{4.5}{D}\right)*FM*(1-0.0032A)*CM$$
where *H* is the horizontal median distance between the center of the palm and the two ankle joints at the beginning or end of the lifting movement (cm). *V* is the vertical distance between the palm and the ground at the beginning or end of the lifting movement (cm). *D* is the vertical distance between the start and end of the move (cm). *FM* is moving frequency parameters, as shown in Table 2 (depending on frequency to determine different coefficients); A is a deviation from the angle (degree) of the sagittal plane; *CM* is the parameters (determined by the difficulty, raise the difficulty to 1, 0.95 and 0.90 from big to small).

It needs to be explained that Asian people of the same weight have a lower lifting capacity than European and American people [34], so it is necessary to use Formula (14) adapted to Chinese people when conducting this study [26]. According to the manual operation of ergonomics in the Chinese National Standards, Part 1: Lifting and Transfer (GB/T310020.1–2014), the load constant in the lifting formula is usually reduced by 15%, that is, 23 kg is reduced to 20 kg, height coefficient reference adjusted from 75 to 72 cm [35]. After many measurements and comparison of the body action size, *H* = 25 cm [18]. The bending waist of this article is all bent within the sagittal plane rather than the twist angle of the body, so A = 0. The moving frequency was studied as a single operation, so the minimum value was 0.2 times per minute, and the duration was less than 1 h, so *FM* = 1.00 by Table 2. According to the body size of the male body of age range from 18 to 60 in Chinese adult human body size (GB10000-88), the fifty percent of the hand function height is 741 mm = 74 cm, which indicates the specific location. This, minus *V* (the vertical distance between the palm and the ground at the beginning or end of the lifting movement), is equal to *D* (the vertical distance between the start and end of the move), that is *D*= (74 − *V*) cm. Moreover, it is difficult to grasp the object in the actual manual lifting task, so *CM* = 0.90. The above data are calculated as follows
(20)$$RWL=20*(1-0.003\left|V-72\right|)*\left(0.82+\frac{4.5}{74-V}\right)*0.90$$

Using *h* instead of *V* to indicate the height of the moving object from the ground, the relationship between the height of the object and the *RWL* is obtained as follows
(21)$$RWL=20*(1-0.003\left|h-72\right|)*\left(0.82+\frac{4.5}{74-h}\right)*0.90$$

#### 2.2.3. Lifting Index (LI)

As mentioned above, LI means the ratio of actual lifting weight and *RWL*. If the LI value is greater than 1.0, that is, the weight of the actual moving object is greater than the *RWL*, the current working state may pose a hazard to the workers. If the LI is greater than 3.0, the damage is significantly increased.

### 2.3. Validation

#### 2.3.1. Parameters Validation

The most vulnerable part of the body is L5-S1 during the process of lifting at the waist, and the spine of the L5-S1 region of the operation will produce considerable pressure stress. NIOSH built a static biomechanical model in the study, which claimed that the force of L5-S1 less than 3400 N does not cause lumbar injuries. If the force reached 3400 N, it would increase the risk of the of the lumbar injury [18]. This section will calculate whether the force of L5-S1 is greater than 3400 N when the comfort is 0, and judge the rationality of the model. If the comfort is *C* ≤ 0, the force of L5-S1 is larger than 3400 N, and at *C* = 0, the force of L5-S1 is close to 3400 N, which indicates that the comfort model is accurate in the theoretical calculation. If the force of L5-S1 is rather less than 3400 N at *C* = 0, the worker can accept a greater load in the lifting task and the value of *RWL* is too small, which shows a waste of manpower. F more than 3400 N shows that an acceptable job load will cause harm to the human body, and the models in these two cases have errors and cannot be used. According to the five formulas listed below, the force *N* (unit: N) of L5-S1 is divided into positive stress (unit: N) and shear stress tau (unit: N) and the Figure 1d can be obtained.
(22)$$F_{1}+F_{2}+F_{3}=0.47M$$
(23)$$\sum F_{X}=0,\;\sum F_{Y}=0$$
(24)σ=F+(0.47M+G)⋅cosα
(25)τ=(0.47M+G)⋅sinα
(26)$$N=\sqrt{\sigma^{2}+\tau^{2}}$$

#### 2.3.2. Simulate Manual Lifting via Jack Software

The Jack (version 8.3; The Siemens Product Lifecycle Management Software Inc., California, CA, USA) used in this study is a human body simulation and ergonomics evaluation software to help organizations in various industries to improve the ergonomic factors of product design task design of the workshop. Jack was originally developed by the Center for Human Modeling and Simulation at the University of Pennsylvania. The main advantage of Jack is its flexible, realistic and detailed three-dimensional human model, especially the models of the hand, spine, shoulder and other parts. The digital human model in the software is composed of 71 segments and 68 joints, and we can also customize the dimensions of the digital human body [36]. Jack supports the professional ergonomics evaluation system, which can statistic analyze physiological parameters like the posture, force and fatigue parameters through a virtual environment. Then, combined with the widely used ergonomics standards in the industry, the interpersonal ergonomics can be analyzed and evaluated to predict and the possible damage risk for the human body. Jack software will be used to simulate the lifting process of carrying *RWL* at different heights, that is, in the condition: *C* = 0 to verify the waist comfort model. The operation of Jack can be roughly divided into five steps: building a virtual environment, creating a virtual human body, defining the size and shape of the body, placing the body in the environment, and assigning tasks to the human body.

#### 2.3.3. Establish a Human Body Model

The default human database of Jack is the ANSUR 1988 human database. Since human body data vary among different nations, this paper needs to establish a Chinese human body model by custom. The data come from GB10000-88, so it can also select the 50 hundred quantiles directly, showing that the height is 167.8 cm, and the weight is 59.0 kg. The advanced proportions panel in Jack software allows users to control human dimensions by making human body measurements and allow themselves to use specific human dimensions to create a human body model. After establishing the human body model, the software will automatically generate the human body model.

#### 2.3.4. Lifting Simulation

Jack software provides a tool for the analysis of the low back spine. However, only the L4-L5 force analysis module is available in Jack. Although this study was conducted for L5-S1, for the following two reasons, we still believe that the Jack software can help our research. First, L4-L5 and L5-S1 are very close to each other, they have similar biomechanical effects, that the forces they receive will not differ much [37]. Secondly, existing studies have found that in exactly the same lifting activities, the force of L5–S1 is greater than L4–L5 [38], so, in the case of *C* = 0, the force of L4–L5 is less than or equal to 3400 N, the pressure at L5-S1 must also be less than 3400 N, indicating that the pressure is acceptable. This section will simulate the lifting process when the comfort level is 0 to judge whether the force of L5–S1 is greater than 3400 N. In the process of simulation lifting, it is necessary to set up the objects with different heights of h and set up the corresponding weight of *RWL*.

## 3. Results

### 3.1. The Result of RWL

The research results here are a summary of *RWL* calculations. The calculation is based on the Formulas (20) and (21), constructed in the research method.

According to the above calculation results, in the range of 28.6 cm ≤ *h* < 74.0 cm, the height of the data person *H* = 168 cm and the weight *M* = 578 N are brought into the Formula (20), and the drawn image is shown as Figure 2a.

Bring the *RWL* corresponding to each lifting operation height *h* in Figure 2a into Formula (14), calculate the value of *F* (*RWL*) about *h*, and draw a trend image, as shown in Figure 2b.

We can find from the Figure 2a that with the increase of the height of the work, *RWL* is increasing, and it grows steadily in the first half of the period and increases exponentially when it is close to the 74 cm limit.

Figure 2b mean that *F* (*RWL*) produced by the vertical ridge muscle decreases as the height of lifting increases. The largest value is 3273.82 N when the height is 34 cm, and the minimum is 2025.69 N when the height is 72 cm.

### 3.2. The Result of the Relationship Between Comfort Degree and RWL, and LI

As shown in Figure 2 and the Formulas (20) and (21) related to *RWL*, when the height *h* is fixed and *RWL* is a fixed value, so the tensile force *F* (*RWL*) of the vertical ridge muscle is also a constant. Therefore, the following conclusion can be made.

*G* = 0 means there is no workload, LI = 0.0, and the pull *F* of the vertical ridge muscle is the minimum value: *Fmin* = 3.42M × sinα. It is known by Formula (18) that *μC* is the minimum value: *μC_min_* = 3.42*M* × sin*α*/*F*(*RWL*), *C_max_* = 1 − *μC_min_* = 1 − 3.42*M* × sin*α*/*F* (*RWL*), the comfort is maximum and *C* < 1.

When *G* = *RWL*, *F* = *F* (*RWL*), *μC* = 1, *C* = 0, LI = 1.0, according to the definition of *RWL* by NIOSH, health workers can work well enough for a long enough time without causing the risk of lower back pain, so the load of *C* = 0 is defined as the extreme lifting load.

When *G* < *RWL*, *F* < *F* (*RWL*), *μC* < 1, 0 < *C* < 1, LI < 1.0, workers can work properly for a long enough time, so the load of 0 < *C*<1 is defined as acceptable lifting load.

When *G* > *RWL*, *F* > *F* (*RWL*), *μC* > 1, *C* < 0, LI > 1.0, at this time the current workload may pose a hazard to the workers, so the load of *C* < 0 is defined as a dangerous lifting load.

According to the lifting index (LI), the gravity of the object in the handling operation is brought into the Formula (14), the corresponding Erector Spinae force *F* is calculated, and the trend image is drawn as shown in Figure 3.

When the lifting index LI is between 0.0 and 3.0, the F of the erector muscle increases by about 1300 N if LI increases by 1.0. The curve LI = 1.0 indicates the comfort *C* = 0 and its lifting weight belongs to the limit load; the part above curve LI = 1.0 means the comfort degree *C* < 0, the moving weight belongs to the dangerous workload; the following part of the curve LI = 1.0 shows the comfort degree 0 < *C* < 1, the carrying weight belongs to the acceptable lifting load.

### 3.3. The Result of Parameters Validation

According to the methods and formulas mentioned in the parameter verification section, we set *G* = *RWL*, and the force of L5-S1 is the pressure value of comfort *C* = 0. The height and weight of the data are *H* = 168 cm and *M* = 578 N, and the figures of normal stress *σ*, shear stress *τ* and resultant force *N* are as Figure 4, Figure 5, Figure 6 respectively.

According to the above calculation and the figures, the stress *N* of L5-S1 is mainly derived from normal stress in the handling operation, and the primary source of the positive stress is the tension F provided by the vertical ridge muscle. When the height of the processing operation is 70 cm, the pressure *N* of L5-S1 is minimum, it is 2652.65 N and the corresponding lifting proposal is 34.80 Kg. At this operating height, the worker’s force of waist is relatively least. In the case of comfort *C* = 0, the pressure of L5-S1 is less than 3400 N, the force is relatively small, and the waist is very comfortable. The pressure and comfort of L5-S1 corresponding to different lifting index LI is calculated according to Formula (18), as shown in Figure 7 and Figure 8.

It can be seen from Figure 7 and Figure 8 that when the comfort level is *C* > 0, the lifting index is 0.0 < LI < 1.0, and the pressure of L5-S1 is less than 3400 N currently, which suggests an acceptable workload. When *C* < 0, the pressure of L5-S1 and the value of *N* is greater than 3400 N. When the lifting index is LI > 2.0, all the values of L5-S1 are greater than 3400 N, which is consistent with the conclusion of the previous experiments. The testing process based on NIOSH that has been widely applied, which is a validated proof. However, in the actual situation, the ability to withstand pressure is different among people with individual physical differences (e.g., age, gender, health condition). This section shows only the biomechanical aspects but lacks values of physical limits, psychophysics limits and comprehensive limits of the validation [39]. Therefore, the test results have some limitations and Jack software will be used to verify the model further.

### 3.4. The Results of Jack Software Simulation

Table 3 shows that during the simulation, the forces of L4-L5 are all less than 3400 N. The force at the L4-L5 lumbar spine increases gradually with the increase in the gravity center of the moving object, which is somewhat different from the force change at the L5-S1 lumbar spine in Figure 6. Overall, the force of L5-S1 is less than that of L4-L5. We use Jack software to simulate the lifting process of waist comfort *C* = 0 and show that the force of L4-L5 is less than 3400 N, thus we have reason to believe that, under the same conditions, the force of L5-S1 is less than the limit value of 3400 N.

## 4. Discussion

A detailed knowledge of lumbar spinal loads is a basic requirement for proper management of various spinal disorders, and effective injury prevention in the workplace, in sports, rehabilitation and computer simulations [40]. Biomechanical models are often used to quantify job risk by estimating waist muscle forces [7]. The study established a static waist comfort model for Chinese men, which can relatively quantitatively calculate the waist force when lifting heavy objects, predict comfort, and enhance the ability to identify risk factors for LBP. This finding allows employers to design or assign tasks based on ergonomics while considering personal physical characteristics. One of the advantages of biomechanical posture analysis is that it can objectively assess the risk of lower back pain, which can be regarded as a prerequisite for dynamic detection of waist force. After all, the objective spinal load measured by the biomechanical model or muscle sensors is not always consistent with the subjective received load, because there are differences in the tolerable load and muscle power among various individuals. Hence, the comfort level can be regarded as a reliable indicator to estimate work-related risk subjects to low back injuries. Likewise, the quantitative measurement of comfort provides a subjective evaluation indicator to design wearable exoskeletons, which is being widely exploring currently. It was noted that this paper developed an understanding of subjective response of waist load which is not merely limited to self-assessment like questionnaires or surveys, and the comfort research can also be extended to other parts of the body, such as common disorders (e.g., neck pain or shoulder pain).

However, the present study contained limitations, including the fact that comfort is a subjective feeling of human being, and it could be affected for a variety of reasons, like physical, physiological and psychological factors. This article only deals with one aspect of physiology in muscle stress without consideration of the influence of other factors on comfort. In the process of establishing a model, the simplified modeling method is used to analyze waist force, and the parameters of the human body selected only reference some national standards. Data sources used in this study are not the latest, and with the improvement in people’s living standards, the size of people’s bodies may be different from the data measured at the time. In this paper, another limitation associated with data is that only male data are selected for analysis, not including female workers, and therefore it cannot be used to account for human specificity. The waist-related research about women is similarly worth studying because the prevalence of LBP in women is higher than in men [41], and the withstand load is lower when confronting the same task [42].

In addition, manual lifting belongs to a dynamic process, and low-back-load people suffered changes over time. Static force analysis is adopted in the current research rather than dynamic detection, so dynamic waist force and comfort level estimation with high accuracy should be investigated in future studies. Lifting can be divided into upward moving and downward moving. However, this paper only considered lifting that moves from down to up, so the waist force model and waist comfort model are limited to be applicable to only the upward moving operation. Furthermore, the bending posture was assumed as sagittal plane bending without considering torsion and other conditions, so the application of the model has a certain scope. Various posture analyses combined with torsion, and 3D motion analysis of human body have already been carried out, with the largest challenge of improving detection accuracy [43].

In the verification process of the model, only the stress of L5-S1, which should be less than 3400 N, is verified, and the forces of other parts and joints of the human body are not calculated. If possible, it is ideal to utilize muscle motion sensors and conduct an experiment to examine the waist force model, and to combine self-reports to test the reliability of the comfort model. Additionally, the psychological endurance of the workers to the workload and the influencing factors of the environment should be considered in further research.

## 5. Conclusions

This research is based on the classic biomechanical formula and has been modified reasonably to adapt to Chinese male body data. Using human posture analysis (HPA) to sort out and derive a large number of waist force formulas and analyze the waist strength under different manual material lifting conditions. An adjustment to the *RWL* formula recommended by NIOSH, and the biomechanical waist comfort model for manual material lifting was established, which was simulated and verified by Jack simulation software. The established waist comfort model can objectively and quickly assess waist force and enhance the ability to identify risk factors for LBP. Although the model can only analyze and calculate static operations, it can provide process and methodological references for subsequent dynamic monitoring of waist force and research for specific populations. Combining muscle sensors such as EMG and LMM in follow-up research can form more comprehensive and scientific research results.

## Figures and Tables

**Figure 1 ijerph-17-05948-f001:**
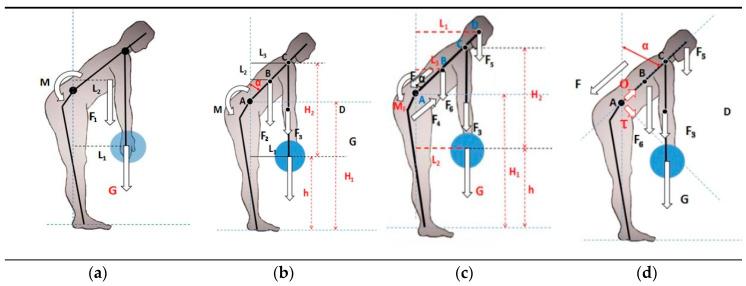
The Simplified L5-S1 Force Model. (**a**) The relationship model between Erector Spinae tension and object gravity; (**b**) The relationship model between Erector Spinae tension and the height of objects from the ground; (**c**) Static balance model when bending over; (**d**) Force analysis of L5/S1 in handling.

**Figure 2 ijerph-17-05948-f002:**
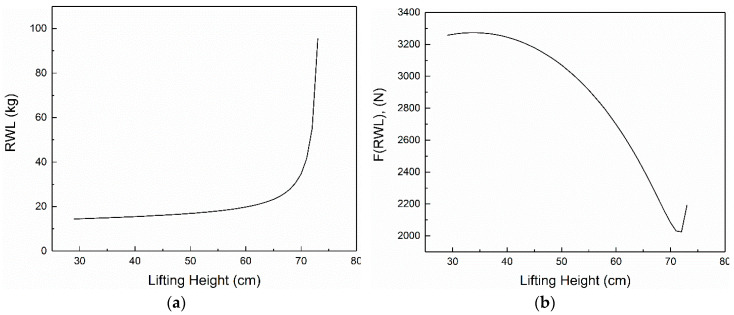
The relationship between *RWL*, *F* (*RWL*) and *h*. (**a**) The relationship between *RWL* and *h*; (**b**) The relationship between *F* (*RWL*) and *h.*

**Figure 3 ijerph-17-05948-f003:**
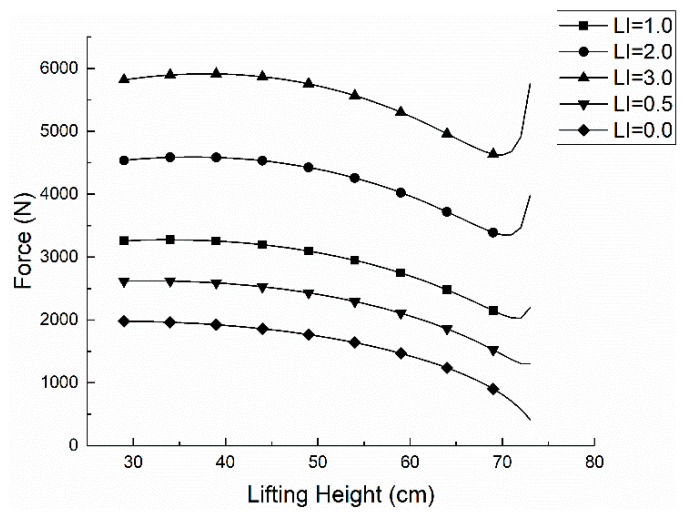
The relationship between *F* and *h*.

**Figure 4 ijerph-17-05948-f004:**
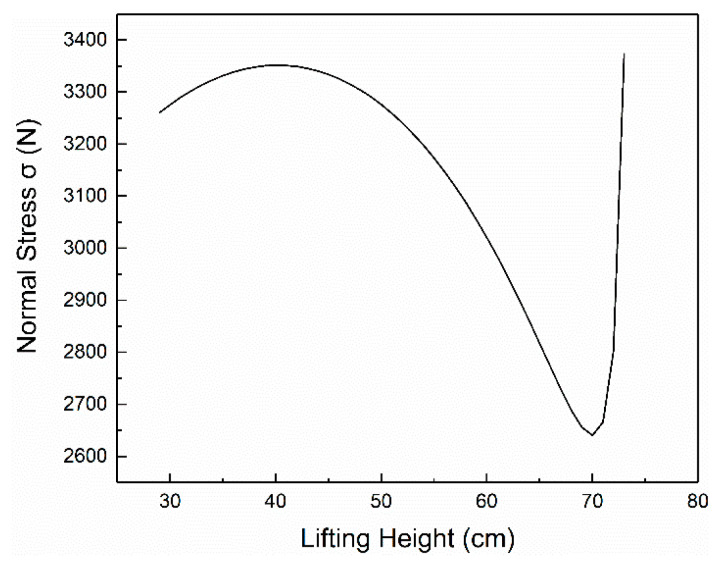
The relationship between *σ* and *h*.

**Figure 5 ijerph-17-05948-f005:**
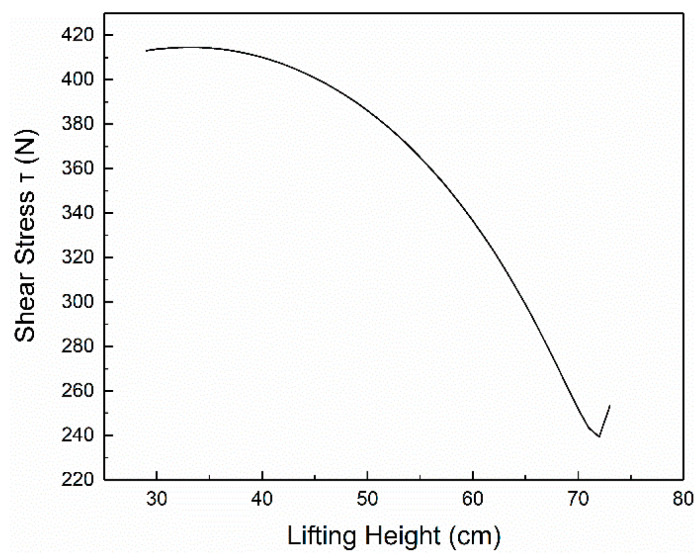
The relationship between *τ* and *h*.

**Figure 6 ijerph-17-05948-f006:**
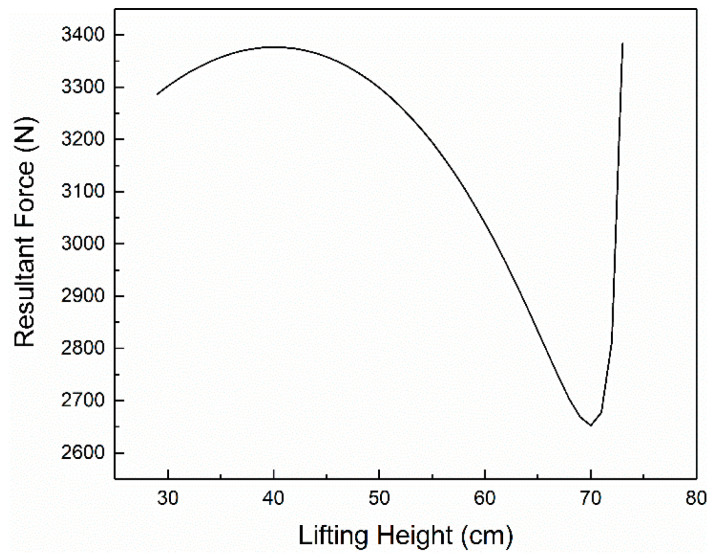
The relationship between *N* of L5-S1 and *h*.

**Figure 7 ijerph-17-05948-f007:**
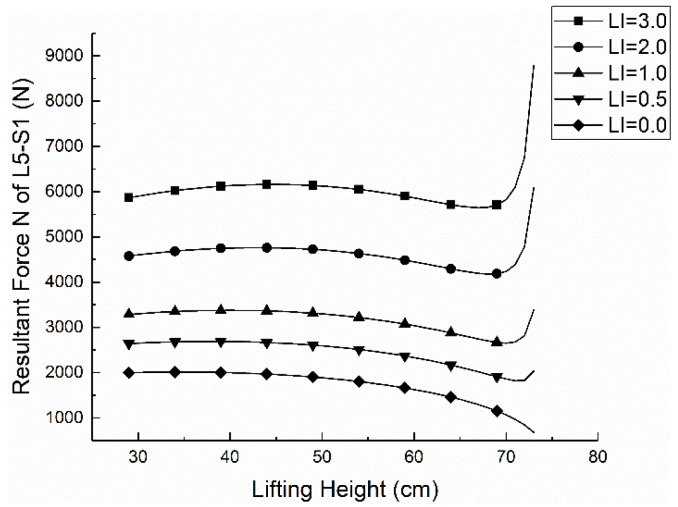
The relationship between *N* of L5-S1 and *h*.

**Figure 8 ijerph-17-05948-f008:**
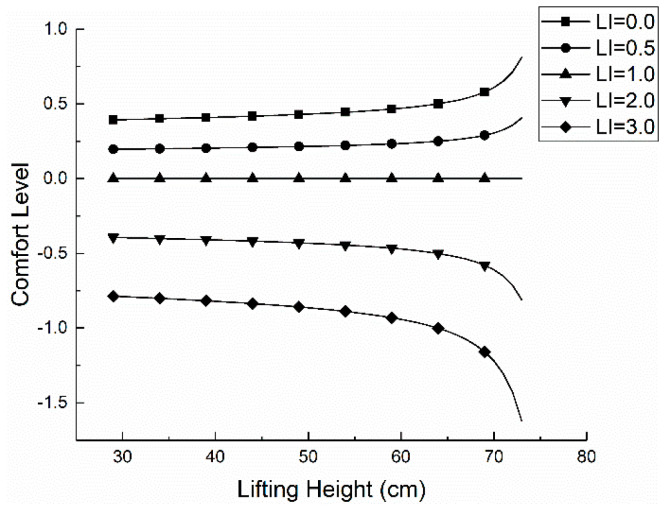
The relationship between *C* and *h*.

**Table 1 ijerph-17-05948-t001:** Male part data in adult human body inertia parameters.

Somatic Segment Name	Centroid Measurement Starting Point	Centroid Position (Unit: mm)	Centroid Relative Position (unit: %)	Relative Mass (unit: %)
Neck	vertex	117.8	46.9	8.62
Upper trunk	Cervical vertebra	115.6	53.6	16.82
Lower trunk	Lower thoracic point	177.8	40.3	27.23
Thigh	Tibial point	254.5	45.3	14.19
A lower leg	Medial malleolus point	224.1	39.3	3.67
Upper arm	Bony point	163.3	47.8	2.43
Forearm	Point of the styloid process of the bone	136.6	42.4	1.25
Hand	Fingertip of middle finger	114.2	36.6	0.64
Foot	Plantar	38.2	48.6	1.48
Whole	Top	734.2	43.8	-

Note 1: the centroid position (M.C.) is determined by the distance from the measuring point to the centroid of the body. Note 2: the average value of the table is the average age of 11,164 adult men (18 years and 60 years old). Note 3: the relative position of the center of mass is the percentage of the upper part of the center of mass in each body segment. Example: the relative position of the center of mass of the head and neck = (head to center mass to head distance head neck length) × 100%; Relative weight of head and neck = head neck mass/overall quality) × 100%.

**Table 2 ijerph-17-05948-t002:** FM frequency factor.

Working Hours						
Frequency F	≤1 h		≤2 h		≤8 h	
(Times/Points)	V < 75 cm	V ≥ 75 cm	V < 75 cm	V ≥ 75 cm	V < 75 cm	V ≥ 75 cm
0.2	1.00	1.00	0.95	0.95	0.85	0.85
0.5	0.97	0.97	0.92	0.92	0.81	0.81
1	0.94	0.94	0.88	0.88	0.75	0.75
2	0.91	0.91	0.84	0.84	0.66	0.66
3	0.88	0.88	0.79	0.79	0.55	0.55
4	0.84	0.84	0.72	0.72	0.45	0.45

Note: F is the frequency of lifting, expressed by the number of times per minute.

**Table 3 ijerph-17-05948-t003:** L4-L5 stress in the software simulation process.

Center of Gravity away from Ground Height (cm)	Mass of Objects(kg)	L4-L5 Force(N)
35	14.97	3143
38	15.27	3120
41	15.61	3148
44	15.99	3179
47	16.43	3213
50	16.94	3218
53	17.56	3226
56	18.34	3235
59	19.37	3247
62	20.86	3268
65	23.26	3308
68	27.92	3388

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
