# Peer review of "A Biomechanical Waist Comfort Model for Manual Material Lifting"

_ijerph, 2020, doi:10.3390/ijerph17165948_

Round 1

Reviewer 1 Report

This is a well-written manuscript, examining the waist comfort model for Chinese men using verified Jack simulation.  Further studies including the exoskeleton will verify these results.

The authors have already addressed some limitations in the discussion. However, I recommend further discussion regarding the practical relevance of the findings gained. In particular, the limitation regarding the combined movement (lifting and rotating). Can these findings improve the current ergonomic recommendations? Or does it make sense to make an individual work routine recommendation based on the individual anatomic situation?

Reviewer 2 Report

The article is written in English using clear and unambiguous text. The article is include sufficient introduction and background to demonstrate how the work fits into the broader field of knowledge. Relevant prior literature is appropriately referenced. The structure of the article is conform to an acceptable format. The aims is appropriated and clearly define the research question, is relevant and meaningful. The investigation is have been conducted rigorously and to a high technical standard. The data is robust, statistically sound, and controlled. The conclusions is appropriately stated, is connected to the original question investigated, and is limited to those supported by the results.

Reviewer 3 Report

Manuscript ID: IJERPH-876710

Manuscript title: A Biomechanical Waist Comfort Model for Manual Material Lifting

Major comments

  1. Introduction, lines 58-60. Reference [5] cannot be used to support the information that physical occupational exposure is the leading risk factor for low back pain. As a matter of fact, most cases of low back pain are nonspecific (https://doi.org/10.1016/s0140-6736(16)30970-9), and psychosocial factors also play a role in low back pain. The issue of improper use of citations is also apparent in the next sentence regarding ref [6]. Please revise the text and provide supporting references.

  1. Introduction, lines 92-96. Reference [17] is a cross-sectional study, which allows to conclude about risks, but not causal relationships such as the one mentioned: ‘when the LI is greater than 3, the extent of the damage will be significantly increased [17]’. Please rephrase to account for the study design and specific outcomes.

  1. Introduction, lines 129-132. What is meant by ‘evidence about how to evaluate career risk’? Doesn’t the proposed model require a longitudinal study for this aim?

  1. Methodology, line 151. The Eretor Spinae is a set of muscles, not only the sacral spinalis.

  1. Methodology, lines 153-154. The L5-S1 lumbar disc does not rest between the fifth segment of the lumbar spine and the first

segment of the tibia.

  1. Methodology. Equations appears misconfigured in the PDF file, could not check them and most parts of the related text.

  1. Validation, lines 33-34 (after Figure 6). Avoid using phrases such as ‘which does not cause lumbar injuries’ as neither your data nor previous literature support them.

  1. Results and Discussion, lines 75-77. It is unclear whether reference [35] specifically mentions the waist comfort model as a basic requirement for proper management of various spine disorders, effective prevention of injuries in the workplace, sports and rehabilitation, and human trials and computer simulations.

  1. Results and Discussion, lines 81-82. I failed to identify new data to support the claim that ‘the proposed comfort computed approach quantitatively reflects people’s response and feelings of load on the waist’.

  1. Conclusions. This section summarizes other parts of the manuscript that could be omitted. Consider focusing on the novelty and applicability of your findings.

Round 2

Reviewer 3 Report

Thank you for your responses. The manuscript is substantially improved from the original version. I have no further suggestions.